# SIM2REAL-HOI: SIM-TO-REAL HOI VIDEO GENERATION VIA DECOUPLED MOTION–APPEARANCE DIFFUSION

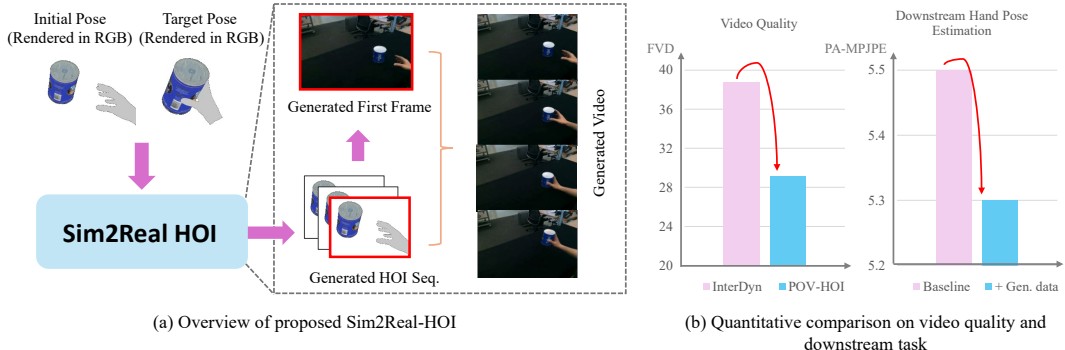

(a) Overview of proposed Sim2Real-HOI

(b) Quantitative comparison on video quality and downstream task

Figure 1: (a) **Overview of proposed Sim2Real-HOI**. Given the initial and desired final poses of both the hand and object, along with the object mesh, Sim2Real-HOI generates photo-realistic in-the-wild videos by decoupling motion and appearance through a diffusion-based generative process, thereby closing the sim-to-real gap without paired real-world supervision. (b) **Evaluation on video quality and downstream task.** Our experiments reveal that Sim2Real-HOI not only generates videos with superior perceptual quality, as evidenced by a lower Fréchet Video Distance (FVD) (Unterthiner et al., 2018) against baselines, but also that these videos serve as effective synthetic data. Incorporating them into training reduces the error of a downstream hand pose estimator, outperforming the model trained solely on real videos.

## ABSTRACT

We present Sim2Real-HOI, a zero-shot framework that closes the sim-to-real gap for hand–object interaction (HOI) video generation using the initial and target poses of both hand and object. Controllable diffusion models like InterDyn and ManiVideo stumble at scale when moving simulation to reality: the quality of generated videos are suboptimal, and they rely on simulator-unobtainable cues such as the first frame. Sim2Real-HOI addresses the problem in two stages: (1) an appearance generator that models both appearance and background using a controllable image diffusion model, and (2) a motion transfer model that transfers motion, generated by a pretrained hand pose generator, to real-world video through a controllable video diffusion model. To improve performance, we incorporate multiple types of conditions that ensure the generated output aligns with the geometry, semantics, and fine details of the hand pose. Extensive experiments on DexYCB and OAKINK2 demonstrate that Sim2Real-HOI enhances the generated quality compared to the best prior work and results in a lower error rate when the generated videos are used to train downstream hand-pose estimators. The code and pre-trained weights will be made publicly available.

## 1 INTRODUCTION

The ability to manipulate objects with two hands represents a fundamental human skill, and the computational understanding of this capability—referred to as Hand-Object Interaction (HOI) un-

| Method / Input Req. | Background | HOI Pose Seq | Object Appearance | Hand Apperance |
|---|:---:|:---:|:---:|:---:|
| CosHand | ✓ | ✓ | ✓ | ✓ |
| InterDyn | ✓ | ✓ | ✓ | ✓ |
| ManiVideo | ✓ | ✓ | ✓ | ✓ |
| Sim2Real-HOI (Ours) | ✗ | ✓* | ✗ | ✗ |

Table 1: Comparison of input requirements for HOI video generation. A checkmark (✓) signifies a condition provided as input to the model, whereas a cross (✗) indicates a condition that is not provided and will be synthesized. The symbol (*) denotes a setting where only the initial and target states are specified.

derstanding—has become increasingly significant in the fields of computer vision and embodied AI. The field has seen a shift towards **data-driven** paradigms, where large-scale HOI datasets are instrumental for accurate hand pose estimation (Zhou et al., 2024; Pavlakos et al., 2024; Dong et al., 2024) and enabling realistic human-to-robot motion transfer (Liu et al., 2025; Lepert et al., 2025; Li et al., 2025). The critical challenge, however, lies in the data itself. Despite considerable investments in collecting real-world HOI sequences with detailed annotations (Liu et al., 2022; Fu et al., 2025; Yang et al., 2022), the reliance on costly and labor-intensive manual labeling poses a fundamental limitation to scalability.

The advent of video diffusion models promises scalable generation of HOI videos. However, state-of-the-art methods (Pang et al., 2025; Akkerman et al., 2025; Sudhakar et al., 2024) critically depend on being conditioned on the first frame of the video, which creates a two-fold problem: (1) a significant *input bottleneck*, as obtaining a first frame that is geometrically consistent with the provided initial hand-object pose sequence is challenging, and (2) a *diversity bottleneck*, as fixing the first frame severely limits the potential for visual randomization, which is essential for data augmentation. Overcoming these bottlenecks by generating realistic videos from minimal inputs constitutes a key unsolved problem.

In this paper, we introduce a pioneering sim-to-real HOI video generation framework that requires *only* the initial and target poses, along with object geometry, as input. By integrating a novel decoupled motion-appearance diffusion process, our method bypasses the need for a conditioned first frame, thereby maximizing both motion and appearance diversity—a capability unattainable by prior work. To ensure high realism, we incorporate multiple conditions that effectively preserve fine hand details. As demonstrated in Table 1 and Figure 1, our approach surpasses existing methods by jointly generating realistic foreground, background, and dynamically interpolated poses.

Our framework comprises two core stages. The **appearance generation** stage synthesizes a realistic initial frame using the controllable image diffusion model Flux (black-forest labs, 2024). This model is conditioned on a fusion of depth maps, semantic masks, and hand keypoint maps, which collectively ensure geometric accuracy, semantic coherence, and the preservation of fine-grained hand details. The **motion generation** stage then animates this frame into a video sequence. We first generate a plausible hand motion trajectory using a pre-trained model, which is subsequently rendered by a controllable video diffusion model (based on CogVideoX (Yang et al., 2024)). Crucially, the video model is conditioned on the same multi-modal inputs to maintain consistency with the generated HOI pose sequence.

We evaluate our method on the DexYCB (Chao et al., 2021) and OAKINK2 (Zhan et al., 2024) benchmarks, where it comprehensively surpasses existing approaches in video generation quality, motion plausibility, and hand pose fidelity. More importantly, as evidenced in Figure 1, the synthesized videos from our method provide substantial value as synthetic data. When used for training, they lead to meaningful gains in the performance of a downstream hand pose estimation model, demonstrating their effectiveness as a data augmentation tool.

We summarize our contributions as follows:

- **Minimal-Conditioning Generation:** We pioneer an HOI video generation framework that requires only sparse pose keyframes and object geometry as input, overcoming the first-frame bottleneck of prior methods.
- **Decoupled Generation Architecture:** We design a novel pipeline that decouples appearance and motion synthesis, leveraging multi-modal conditions to achieve superior realism and diversity.

- **State-of-the-Art Performance and Utility:** Our method achieves superior results on established benchmarks and proves its practical value by enabling significant gains in downstream task performance through effective data augmentation.

## 2 RELATED WORKS

### 2.1 HAND-OBJECT MOTION SYNTHESIS

Synthesizing high-fidelity hand-object motion is a fundamental challenge in computer animation and robotic grasping (Agarwal et al., 2023; Ghosh et al., 2023; Christen et al., 2024). Prevailing data-driven approaches rely on supervised learning from large-scale, well-annotated datasets (Grady et al., 2021; Jiang et al., 2021; Karunratanakul et al., 2020; Dong et al., 2024; Pavlakos et al., 2024; Christen et al., 2022; Liu & Yi, 2024; Li et al., 2025; Zhong et al., 2025; Zhou et al., 2024). However, the scalability of these methods is constrained by their dependence on costly and difficult-to-acquire data (Fan et al., 2023; Hampali et al., 2020; Liu et al., 2022; 2024; Fu et al., 2025; Yang et al., 2022; Zhan et al., 2024; Chao et al., 2021). To circumvent this limitation, reinforcement learning (RL) has emerged as a promising alternative. Methods like (Christen et al., 2024; Xu et al., 2023) generate reference grasps before synthesizing motions, while GraspXL (Zhang et al., 2024a) learns a generalizable grasping policy directly in simulation, eliminating the need for predefined references. These RL-based techniques produce high-quality interaction data, forming a robust foundation for sim-to-real transfer.

### 2.2 CONTROLLABLE VIDEO GENERATION

Recent breakthroughs in video generation foundation models (Yang et al., 2024; Blattmann et al., 2023; Wan et al., 2025; Kong et al., 2024; Agarwal et al., 2025) have intensified interest in controllable generation that precisely aligns with user intent. While text-to-video and image-to-video models (Agarwal et al., 2025; Wan et al., 2025; Yang et al., 2024; Singer et al., 2022; Qing et al., 2024; Guo et al., 2023; Wiersma et al., 2025; Zhang & Agrawala, 2025) have demonstrated impressive capabilities, they often lack the granularity for specialized tasks. This has spurred research into integrating more precise control signals, such as semantic maps, depth, and camera motion. ControlNet (Zhang et al., 2023) and its variants (Gu et al., 2025; Guo et al., 2024b) enable conditioning on dense inputs, while works like VideoComposer (Wang et al., 2023) fuse multiple conditions for enhanced control. Camera motion has been explicitly modeled by embedding parameters into diffusion models (He et al., 2024; Bai et al., 2025). However, generating videos of hand-object interactions (HOI) presents a unique challenge due to the high degrees of freedom in hand motion. This demands even more enriched and specialized control mechanisms—combining semantic, geometric, and precise pose cues—to achieve the necessary fidelity and accuracy.

### 2.3 HAND-OBJECT INTERACTION IMAGE & VIDEO GENERATION

Generating Hand-Object Interaction (HOI) content is vital for understanding human activities. Prior work on *HOI image generation* (Hu et al., 2022; Kwon et al., 2024; Pelykh et al., 2024; Wang et al., 2025; Ye et al., 2023; Zhang et al., 2024b; Chen et al., 2025) typically conditions on 2D signals like segmentation masks and keypoints. However, these static methods cannot capture the dynamic nature of interactions. Recently, several studies (Sudhakar et al., 2024; Pang et al., 2025; Akkerman et al., 2025; Dang et al., 2025; Ye et al., 2025) have explored *HOI video generation*. InterDyn (Akkerman et al., 2025) conditions on hand mask sequences via ControlNet (Zhang et al., 2023), but under-utilizes the rich conditions available from simulators. ManiVideo (Pang et al., 2025) introduces an occlusion-aware representation but requires human appearance data, which is not available from simulators like GraspXL (Zhang et al., 2024a). More critically, these methods primarily focus on generation quality and have not thoroughly investigated the *downstream utility* of their synthesized data, which is essential for validating practical impact beyond perceptual metrics.

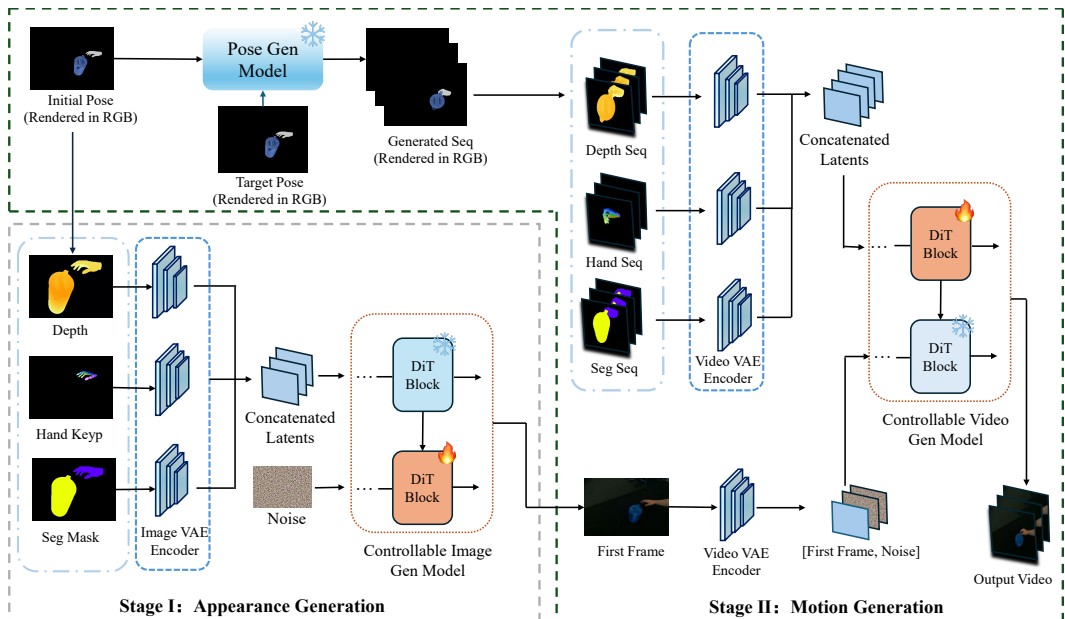

Figure 2: **Overview of our two-stage generation pipeline.** (1) **Appearance Generation:** A controllable image diffusion model synthesizes the first frame from multi-modal conditions (depth, semantic masks, keypoints). (2) **Motion Generation:** A pre-trained hand pose generator produces intermediate poses, which are then rendered into a full video sequence by a video diffusion model, conditioned on the same controls as appearance generation.

## 3 THE PROPOSED METHOD

### 3.1 OVERVIEW

Figure 2 illustrates Sim2Real-HOI. Conditioned on an initial MANO (Romero et al., 2022) hand pose $\mathbf{h}_0 \in \mathbb{R}^{51 \times 3}$, an object mesh $\mathbf{m}$ without appearance, an initial 6-DoF object pose $\mathbf{o}_0 \in \mathbb{R}^6$, and a target hand pose $\mathbf{h}_T \in \mathbb{R}^{51 \times 3}$, our generative model

$$f_{\boldsymbol{\theta}} : (\mathbf{h}_0, \mathbf{m}, \mathbf{o}_0, \mathbf{h}_T) \; \rightarrow \; \{I_t\}_{t=0}^T \tag{1}$$

produces a photo-realistic video that (i) begins with $\mathbf{h}_0$, (ii) ends with $\mathbf{h}_T$, and (iii) depicts a temporally-coherent grasp-to-place motion. All hand poses are parameterised by global translation + rotation plus joint angles; frames $I_t$ are RGB images.

Jointly modelling appearance and motion is notoriously hard because of the high-dimensional spatio-temporal manifold (Guo et al., 2024a). We therefore disentangle generation into two stages: (i) **appearance**—a pose-conditioned image diffusion model synthesises the first frame $I_0$ given initial hand–object poses and object mesh $(\mathbf{h}_0, \mathbf{o}_0, \mathbf{m})$ (Sec. 3.2); (ii) **motion**—a pretrained pose generator produces aligned sequences $\{\mathbf{h}_t, \mathbf{o}_t\}_{t=0}^T$, which are injected into a video diffusion model to animate $I_0$ into a photo-realistic clip (Sec. 3.3).

### 3.2 STAGE I: APPEARANCE GENERATION STAGE

**Bridge Conditions for Sim-to-Real HOI Video Synthesis** The primary objective of this work is to enhance the visual quality of simulated videos while preserving other conditions, thereby bridging the gap between simulation and real-world scenarios. By incorporating both geometric information (e.g., depth maps) and semantic data (e.g., segmentation masks) from the simulator, we seek to accurately reconstruct the visual representation of scenes and objects, while ensuring consistency across all other conditions. However, relying solely on these two data types proves insufficient for accurately generating Hand-Object Interactions (HOI) images or videos. This limitation stems from the complexity and high degree of freedom inherent in hand movements, which cannot be fully captured by geometric and semantic data alone. Specifically, these conditions fail to account for critical

details, such as the number of fingers and their individual poses. To address this challenge, we introduce an additional condition—hand keypoint sequences, as proposed by (Zhang et al., 2024b)—to enable more precise and accurate hand pose generation. This approach facilitates the generation of realistic hand poses, thereby enhancing the overall realism of the interaction. In section 4.2 and 4.3, we explore the influence of every condition.

We fine-tune Flux (black-forest labs, 2024) with a ControlNet (Zhang et al., 2023) fork that accepts depth $D_0$, segmentation $S_0$ and hand-keypoint image $K_0$ ($H{\times}W{\times}3$ each). All cues are VAE-encoded to $\frac{H}{8} \times \frac{W}{8} \times 16$ latents, concatenated channel-wise and Injected into two layers of DiT (Peebles & Xie, 2023) blocks, with weights initialized from the first two layers of original Flux.:

$$f_l = f_l + \mathcal{Z}(f'_l), \tag{2}$$

where $f_l$ is the output of the $l$-th layer of the original DiT (Peebles & Xie, 2023) blocks, and $f'_l$ is the output of the $l$-th layer of the duplicated DiT blocks whose input is the concentrated conditions. Here, $l \in \{0, 1\}$, and $\mathcal{Z}$ represent the zero-convolution layer, which is a $1 \times 1$ convolution with all parameters initialized to zero. During training, only the parameters of ControlNet are updated.

### 3.3 STAGE II: MOTION GENERATION STAGE

To obtain the target video sequence we cascade a pretrained hand-motion generator with a controllable video diffusion model. As illustrated in Figure 2, GraspXL (Zhang et al., 2024a) consumes the initial MANO hand pose $\mathbf{h}_0$ , the 6-DoF object pose $\mathbf{o}_0$ and object mesh $\mathbf{m}$ to produce aligned trajectories $\{\mathbf{h}_t, \mathbf{o}_t\}_{t=0}^T$. We rasterize depth maps, instance-level segmentation masks, and 2-D hand keypoint images at each frame. The pretrained video VAE encodes these conditions into a latent tensor of shape $\mathbb{R}^{\frac{T+1}{4} \times \frac{H}{8} \times \frac{W}{8} \times 16}$, after which we concentrate these latents and inject them into CogVideo-X through 12 duplicate DiT blocks, as outlined in Eq. 2. During training each cue is randomly masked with probability 0.2 to prevent over-reliance on any single modality.

## 4 EXPERIMENT

### 4.1 EXPERIMENT SETTINGS

**Datasets and Data Processing.** We evaluate our method on two standard benchmarks for HOI video generation: DexYCB (Chao et al., 2021) and OAKINK2 (Zhan et al., 2024). For DexYCB, we adopt the s0-split, comprising 6,400 training and 1,600 validation videos. Due to the scale of OAKINK2, we use a curated subset of 8,000 video clips (each 49 frames long), split into 6,400 for training and 1,600 for validation. The conditions for our model—depth maps, semantic masks, and hand keypoints—are derived as follows: depth maps are estimated using DepthCrafter (Hu et al., 2025), while semantic and keypoint information are obtained directly from the dataset annotations.

**Evaluation Metrics.** We employ a comprehensive set of metrics to evaluate our method from four perspectives:

- **Image Quality:** We assess perceptual quality using Structural Similarity Index Measure (SSIM), Learned Perceptual Image Patch Similarity (LPIPS) (Zhang et al., 2018) and Peak Signal-to-Noise Ratio (PSNR).
- **Spatio-temporal Coherence:** We adopt Fréchet Video Distance (FVD) (Unterthiner et al., 2018) to evaluate overall video realism, using the implementation from (Skorokhodov et al., 2022).
- **Motion Fidelity:** We use the Motion Fidelity (MF) metric (Yatim et al., 2024) to quantify dynamic accuracy. For each video, we sample 100 foreground points (on hands/objects), track them using CoTracker3 (Karaev et al., 2024), and compare the trajectories between generated and ground-truth videos. For a ground-truth tracklet $\mathcal{T} = \{\tau_1, \ldots, \tau_T\}$ and a generated tracklet $\tilde{\mathcal{T}} = \{\tilde{\tau}_1, \ldots, \tilde{\tau}_T\}$ where $\tau_t \in \mathbb{R}^2$, MF is defined as:

$$\text{MF} = \frac{1}{|\tilde{\mathcal{T}}|} \sum_{\tilde{\tau} \in \tilde{\mathcal{T}}} \max_{\tau \in \mathcal{T}} \textbf{corr}(\tau, \tilde{\tau}) + \frac{1}{|\mathcal{T}|} \sum_{\tau \in \mathcal{T}} \max_{\tilde{\tau} \in \tilde{\mathcal{T}}} \textbf{corr}(\tau, \tilde{\tau}). \tag{3}$$

The correlation between two tracks is computed as:

$$\textbf{corr}(\tau, \tilde{\tau}) = \frac{1}{F} \sum_{k=1}^{F} \frac{\mathbf{v}_k \cdot \tilde{\mathbf{v}}_k}{\|\mathbf{v}_k\| \|\tilde{\mathbf{v}}_k\|}, \tag{4}$$

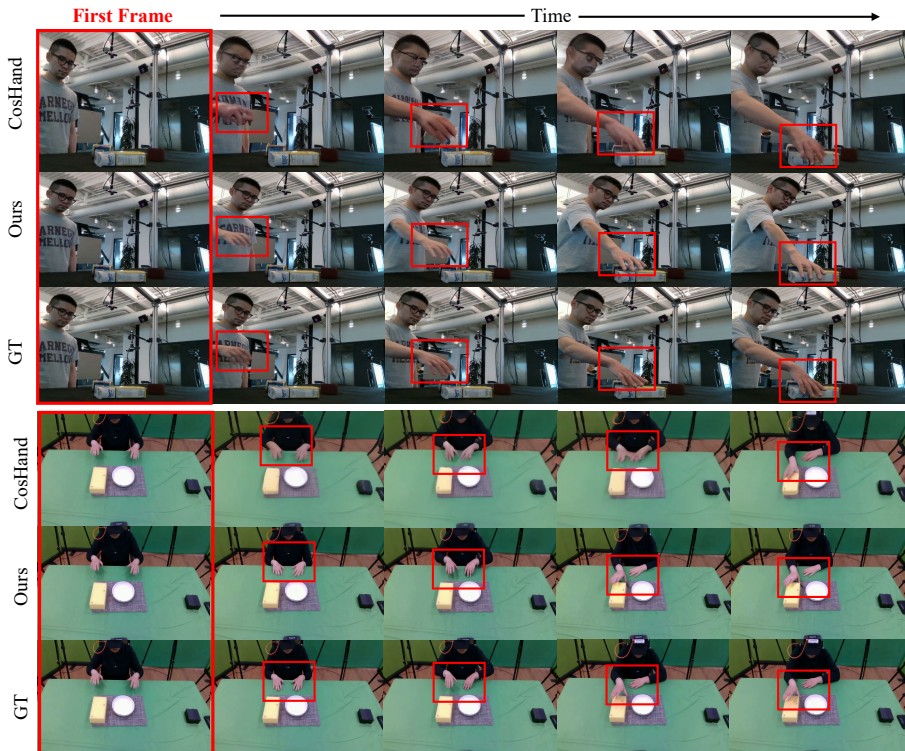

Figure 3: **Qualitative comparison against CosHand.** Example results on DexYCB and OAKINK2 highlight the strengths of our method in two key areas: (1) higher visual fidelity in both foreground and background generation, and (2) improved geometric accuracy of the synthesized hand poses.

| Method | FVD (↓) | MF (↑) | LPIPS (↓) | SSIM (↑) | PSNR (↑) | MPJPE (↓) | Resolution |
|---|---|---|---|---|---|---|---|
| CosHand | 58.51 | 0.591 | 0.139 | 0.767 | 23.20 | 30.05 | 256 x 256 |
| InterDyn | 38.83 | 0.680 | 0.119 | 0.848 | 24.86 | - | 256 x 384 |
| ManiVideo | - | - | 0.079 | 0.913 | 30.10 | 57.30 | - |
| Ours w/ seg | 33.23 | 0.695 | 0.077 | 0.900 | 29.27 | 21.14 | 480 x 720 |
| Ours w/ depth | 30.00 | 0.703 | 0.070 | 0.906 | 29.15 | 23.16 | 480 x 720 |
| Ours w/ hand | 33.41 | **0.713** | 0.086 | 0.901 | 29.07 | 20.70 | 480 x 720 |
| Ours w/ all | **29.13** | 0.712 | **0.069** | **0.914** | **30.17** | **19.37** | 480 x 720 |

Table 2: **Quantitative comparison on DexYCB dataset.** Our method is evaluated against Cos-Hand, InterDyn, and ManiVideo. Results for InterDyn and ManiVideo are taken from their original papers. For fair comparison, CosHand was fine-tuned on the s0-split training set identical to ours. Our approach achieves state-of-the-art performance across all metrics (FVD, LPIPS, MF, MPJPE) while generating high-resolution 480x720 videos.

where $\mathbf{v}_k = (v_k^x, v_k^y)$ and $\tilde{\mathbf{v}}_k = (\tilde{v}_k^x, \tilde{v}_k^y)$ are the displacement vectors at the $k$-th frame for tracks $\tau$ and $\tilde{\tau}$, respectively.

- **Hand Pose Accuracy:** We report Mean Per-Joint Position Error (MPJPE) in millimeters (Fan et al., 2023), measuring the average Euclidean distance between the 21 predicted and ground-truth hand joints after root alignment. Lower MPJPE indicates better pose estimation accuracy.

## 4.2 MAIN RESULTS

**Baselines.** We compare our method against state-of-the-art HOI video generation approaches: ManiVideo (Pang et al., 2025), InterDyn (Akkerman et al., 2025), and CosHand (Sudhakar et al., 2024) on the DexYCB dataset (Chao et al., 2021). For ManiVideo and InterDyn, we report results directly from their original publications (omitting metrics for which results were unavailable due to these methods not being open-source). For CosHand, we use the official implementation and fine-tune it on the DexYCB s0-split training set for a fair comparison. We also evaluate on OAKINK2,

| Method | FVD (↓) | MF (↑) | LPIPS (↓) | SSIM (↑) | PSNR (↑) | MPJPE (↓) |
|--------|---------|--------|-----------|----------|----------|-----------|
| CosHand | 68.76 | 0.651 | 0.156 | 0.765 | 23.84 | 14.49 |
| Ours w/ seg | _48.97_ | _0.708_ | _0.084_ | 0.831 | 25.76 | 9.61 |
| Ours w/ depth | 50.85 | 0.702 | 0.086 | _0.845_ | _26.98_ | 10.07 |
| Ours w/ hand | 52.41 | 0.671 | 0.113 | 0.838 | 25.66 | _8.01_ |
| Ours w/ all | **46.31** | **0.777** | **0.081** | **0.851** | **28.36** | **7.01** |

Table 3: **Quantitative results on the OAKINK2 dataset.** Comparison of our method with Cos-Hand. For a fair evaluation, both models are trained on the same dataset. Our approach achieves state-of-the-art performance, outperforming CosHand across all evaluated metrics.

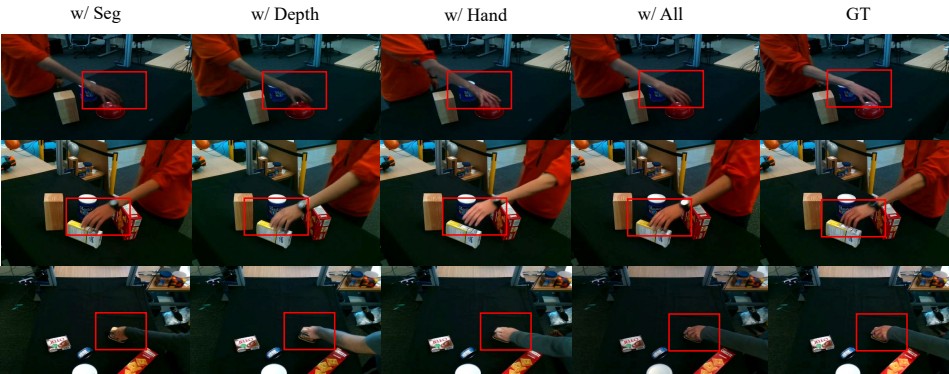

Figure 4: **Ablation study on input conditions on DexYCB dataset.**

comparing against a similarly fine-tuned CosHand model. All baselines are evaluated in an image-to-video setting where the ground-truth first frame is provided, as this is required by these methods.

**Quantitative Comparisons.** Our quantitative evaluation (Tables 2, 3) demonstrates that our method achieves state-of-the-art performance across most metrics. We attribute this superiority to our multi-conditioning strategy, which provides the diffusion model with rich geometric and semantic cues (depth, masks, keypoints) to jointly optimize for visual realism and pose accuracy. In contrast, baseline methods exhibit limitations: InterDyn, ManiVideo, and CosHand rely on more limited conditioning signals or are built upon foundation models that struggle to capture the intricacies of hand-object interactions, leading to suboptimal performance.

**Qualitative Comparisons.** As shown in Figure 3, our method generates visually superior results compared to CosHand, even when CosHand is fine-tuned on the same training data. We identify two primary limitations in CosHand: (1) its reliance on hand masks as the sole conditioning signal provides insufficient geometric guidance for reconstructing precise hand poses, and (2) its lack of explicit temporal modeling mechanisms leads to inconsistent frame-to-frame outputs. In contrast, our approach addresses these issues by leveraging a video diffusion foundation model equipped with temporal attention to enforce coherence across frames. Furthermore, the use of hand keypoint maps as a conditioning signal explicitly preserves the structural details of hand configurations, resulting in more accurate and smooth video sequences.

### 4.3 ABLATION STUDIES ON INPUT CONDITIONS

We conduct an ablation study on the DexYCB dataset to evaluate the contribution of different input conditions. The results (Tables 2, 3, and 4) yield three key observations:

| Method | FVD (↓) | MF (↑) | LPIPS (↓) | SSIM (↑) | PSNR (↑) | MPJPE (↓) |
|--------|---------|--------|-----------|----------|----------|-----------|
| Ours w/o seg | 29.62 | _0.711_ | _0.071_ | 0.899 | 29.95 | 20.46 |
| Ours w/o depth | 29.53 | _0.711_ | 0.073 | 0.902 | 29.57 | _19.92_ |
| Ours w/o hand | _29.32_ | **0.712** | _0.071_ | _0.906_ | **30.60** | 22.51 |
| Ours w/ all | **29.13** | **0.712** | **0.069** | **0.914** | _30.17_ | **19.37** |

Table 4: **Ablation study on input conditions on DexYCB dataset.**

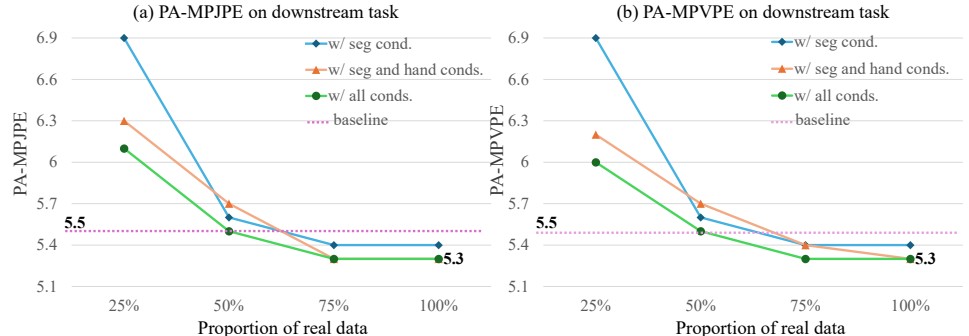

Figure 5: **Sim-to-real transfer results.** Sim2Real-HOI can generate realistic videos given initial and target states.

Figure 6: Data augmentation analysis with varying ratios of real data. We augment different portions of the DexYCB training set (25%, 50%, 75%, 100%) with our generated synthetic data. The **baseline** (dashed line) indicates performance when training solely on 100% of the real DexYCB data without synthetic augmentation.

- The performance improves with an increasing number of conditions, validating the effectiveness of our multi-condition design.
- Even when using the same segmentation mask condition as CosHand and InterDyn, our method achieves superior results, demonstrating the advantage of our pipeline.
- While using only hand keypoints yields low MPJPE (due to explicit pose supervision), it underperforms on other metrics due to the lack of broader geometric and semantic context. This highlights the necessity of combining detailed local cues (keypoints) with global scene understanding (depth, semantics) for optimal performance.

Visual results in Figure 4 further support these findings: using all conditions produces accurate poses; semantic masks or depth maps alone lead to pose inaccuracies; and keypoints alone degrade appearance quality.

### 4.4 SIM-TO-REAL TRANSFER

We conduct a sim-to-real transfer experiment to validate the effectiveness of our full pipeline. We employ GraspXL (Zhang et al., 2024a) as our hand motion generator due to its superior performance and strong generalization. Using objects from the DexYCB dataset, we randomly initialize the hand and object poses along with a target hand pose. GraspXL generates the intermediate motion sequence, which is then used to render the necessary conditions (depth, semantic masks, keypoints) for our video generation model. As shown in Figures 1 and 5, Our method effectively synthesizes realistic videos from minimal input, consisting solely of the initial and target poses, along with the object geometry. This capability stems from our decoupled generation architecture, which effectively integrates the motion prior from GraspXL with the appearance modeling of our diffusion model. The utility of these synthesized videos for downstream tasks is explored in Section 4.5.

### 4.5 DOWNSTREAM TASK VALIDATION

To evaluate the utility of our generated videos, we employ them for data augmentation in a hand pose estimation task. We use SimpleHand (Zhou et al., 2024) as the pose estimation model, which regresses MANO parameters (Romero et al., 2022) from a single image. Our Sim2Real-HOI pipeline,

| Setting | PA-MPJPE (↓) | PA-MPVPE (↓) | F-Score@05 (↑) | F-Score@15 (↑) |
|---|---|---|---|---|
| All real data | 5.5 | 5.5 | 0.7953 | 0.9899 |
| All gen. data | 8.2 | 8.1 | 0.6274 | 0.9626 |
| All gen. + 25% real data | 6.1 | 6.0 | 0.7512 | 0.9851 |
| All gen. + 50% real data | 5.5 | 5.5 | 0.8001 | 0.9879 |
| All gen. + 75% real data | 5.4 | **5.3** | 0.7984 | 0.9899 |
| All gen. + 100% real data | **5.3** | **5.3** | **0.8025** | **0.9904** |

Table 5: Downstream task evaluation on SimpleHand (Zhou et al., 2024).

trained on DexYCB, generates 3,400 video sequences (207,400 frames) for augmentation. We combine this synthetic data with varying subsets (25%, 50%, 75%, 100%) of the original DexYCB s0-split training set (406,888 frames). All models are evaluated on the DexYCB validation set using four metrics: Procrustes-Aligned Mean Per-Joint Position Error (PA-MPJPE), Procrustes-Aligned Mean Per-Vertex Position Error (PA-MPVPE), and F-Score. PA-MPJPE/PA-MPVPE measure the average Euclidean distance (in mm) after Procrustes alignment between the predicted and ground-truth joints/vertices, respectively.

The quantitative results (Table 5) demonstrate that incorporating our generated data consistently improves hand pose estimation accuracy across all metrics. Figure 6 reveals two key trends: (1) model performance improves monotonically with the amount of real data, and (2) most notably, using only 50% of the real data augmented with our synthetic samples achieves competitive performance with the 100% real data baseline. This indicates that our synthetic data can effectively compensate for reduced real data volume. Furthermore, the superior performance achieved using videos generated with multiple conditions validates the importance of our multi-conditioning approach for producing diverse and useful training data.

### 4.6 ZERO-SHOT RESULTS

To evaluate the generalizability of our approach, we test our model trained on the DexYCB dataset (single-hand interactions) directly on the OAKINK2 dataset (bimanual interactions) in a zero-shot setting. As shown in Figure 7, our method generates plausible videos that maintain reasonable alignment with ground-truth hand poses and visual details, despite the significant domain shift. This cross-dataset generalization

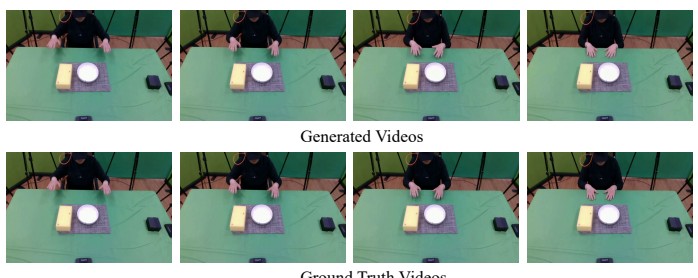

Generated Videos

Ground Truth Videos

Figure 7: Zero-shot result on OAKINK2 dataset. We use the weight trained on DexYCB dataset.

capability can be attributed to our use of pretrained video diffusion model weights as a strong foundation, combined with the ControlNet mechanism (Zhang et al., 2023), which helps preserve the model's original generation quality while adapting to new conditioning signals.

## 5 CONCLUSION

This paper proposed Sim2Real-HOI, a framework that addresses the challenge of generating realistic HOI videos from minimal pose inputs. Our decoupled, multi-condition architecture produces superior results in both perceptual quality and geometric accuracy, and demonstrates practical utility through enhanced downstream task performance. While our method shows strong generalization, future work could explore extending it to more complex object interactions or unifying the motion and appearance stages into an end-to-end model. We believe our contributions provide a solid foundation for future research in generative models for embodied AI.

## ETHICS STATEMENT

This work adheres to the ICLR Code of Ethics. No human subjects or animal experimentation were involved in this study. All datasets used, including those provided by the authors, were sourced in compliance with relevant usage guidelines, ensuring the protection of privacy. We have taken measures to avoid any biases or discriminatory outcomes in our research process. No personally identifiable information was used, and no experiments were conducted that could raise privacy or security concerns. We are committed to maintaining transparency and integrity throughout the research process

## REPRODUCIBILITY STATEMENT

We have made every effort to ensure that the results presented in this paper are reproducible. Our experimental setup is detailed in Section 4.1 and Section A.3. The code is available at https://anonymous.4open.science/r/Sim2Real-HOI-704C/.

Additionally, the public datasets used in this paper, such as DexYCB (Chao et al., 2021) and OAKINK2 (Zhan et al., 2024), are publicly available, ensuring consistent and reproducible evaluation results.

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

## A  APPENDIX

### A.1  USE OF LLMS

We employ GPT-5 to improve and elevate the quality of our written content, primarily for enhancing accuracy and ensuring native-level expression. Its application is focused solely on refining language, rather than idea generation or other functions.

### A.2  TOOLKIT

For access to our fully anonymous code toolkit, please visit: https://anonymous.4open.science/r/Sim2Real-HOI-704C/

### A.3  IMPLEMENTATION DETAILS

**Training Details**: Our model was trained on a setup consisting 8 x NVIDIA 800 GPUs, with a batch size of 4 x 8 and a learning rate of $1 \times 10^{-4}$. The training process involved 8,000 training steps, using the AdamW optimizer and the DeepSpeed training architecture (Rajbhandari et al., 2020).

**Evaluation Details**: For the evaluation of video generation, we sample 1,600 videos, each consisting of 49 frames, from the test set. For the evaluation of Mean Per Joint Position Error (MPJPE), we utilize Hamer (Pavlakos et al., 2024) to estimate the hand joints in the generated videos, and compute the loss by comparing the estimated joint positions with the ground truth hand joints. To assess the performance on downstream tasks, we train the SimpleHand model for 200 epochs using its official implementation.

### A.4  RESULTS

We provide more qualitative results in Figure 8 and Figure 9 for DexYCB dataset, Figure 10 and Figure 11 for OANINK2 dataset and Figure 12 for sim-to-real transfer.

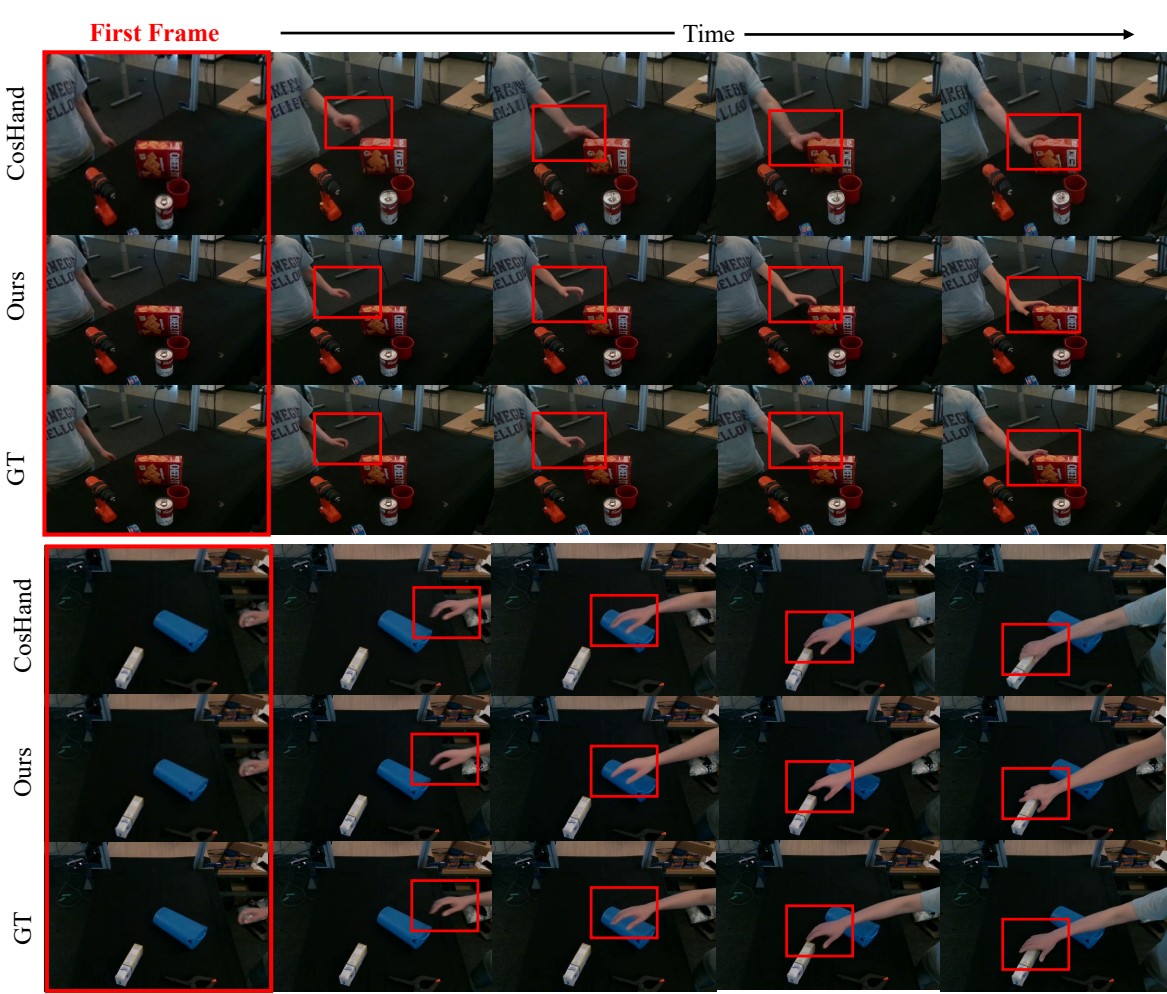

Figure 8: More qualitative results on DexYCB dataset (a).

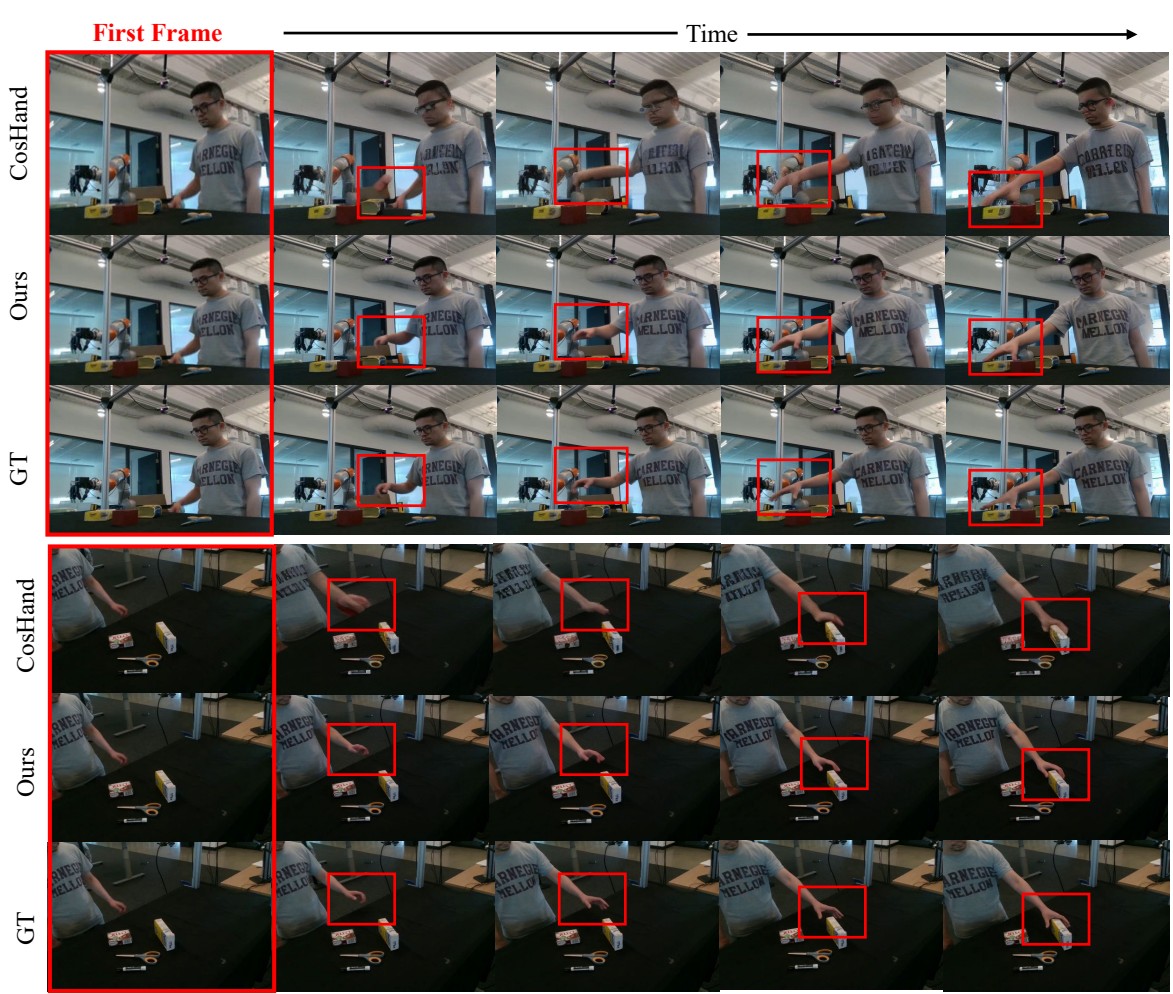

Figure 9: More qualitative results on DexYCB dataset (b).

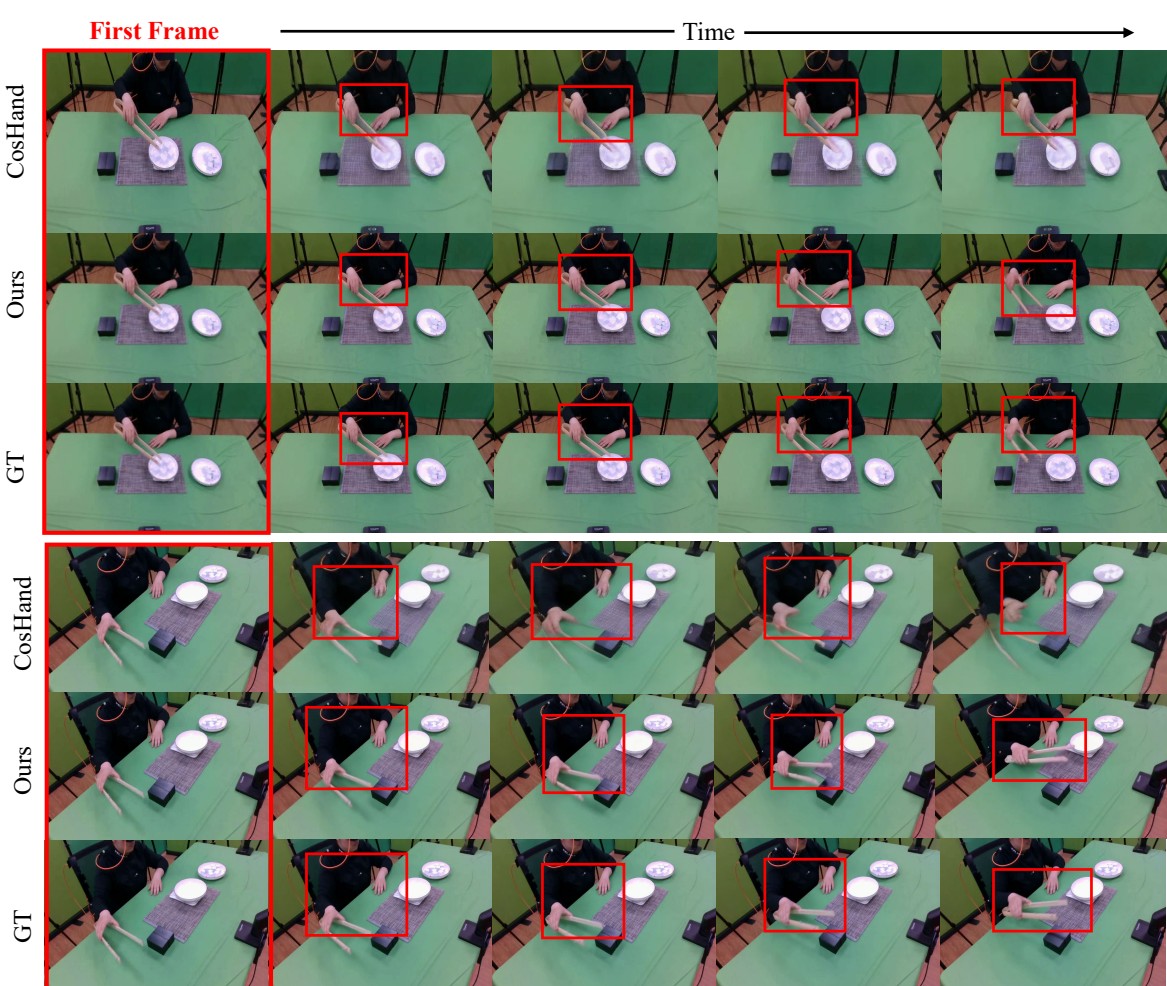

Figure 10: More qualitative results on OAKINK2 dataset (a).

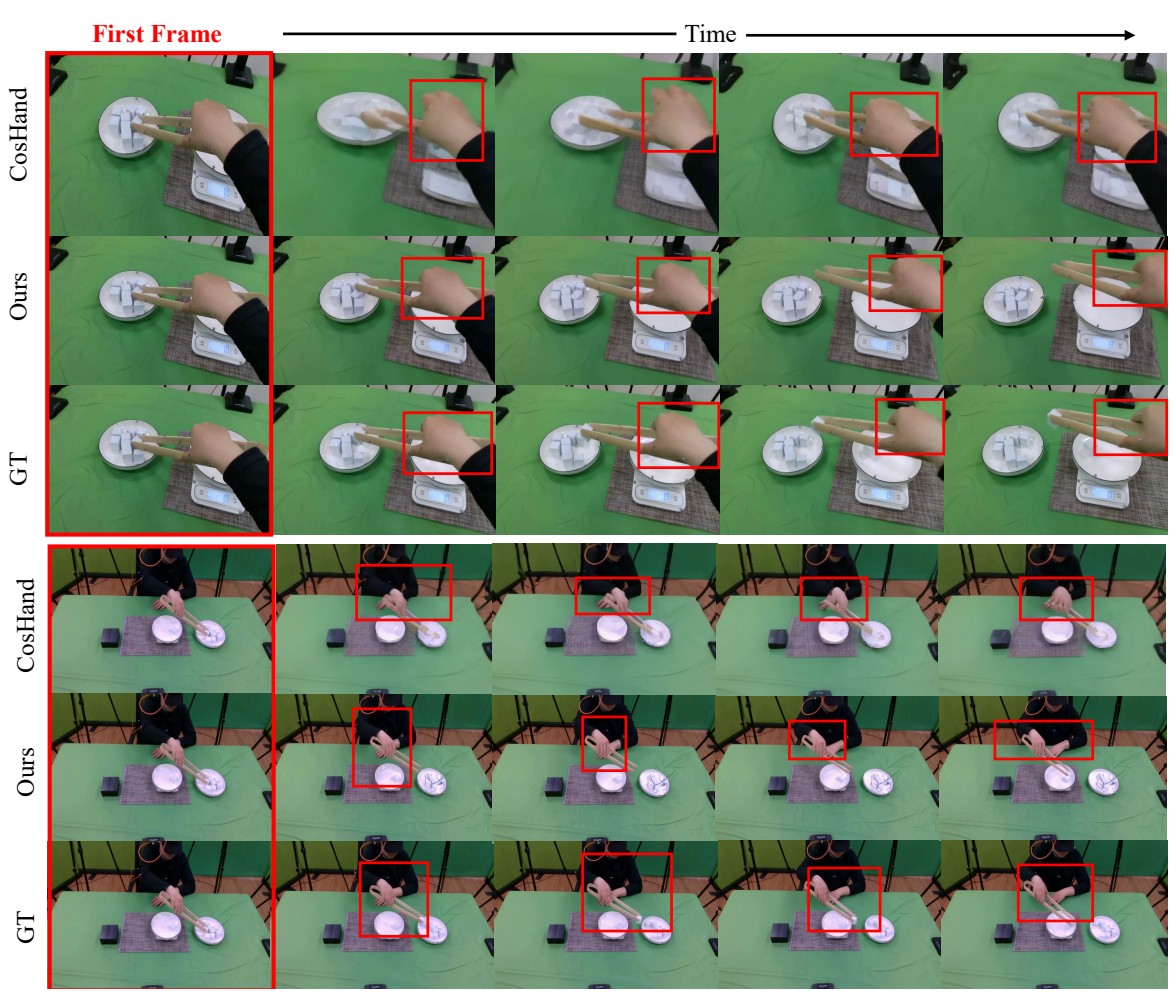

Figure 11: More qualitative results on OAKINK2 dataset (b).

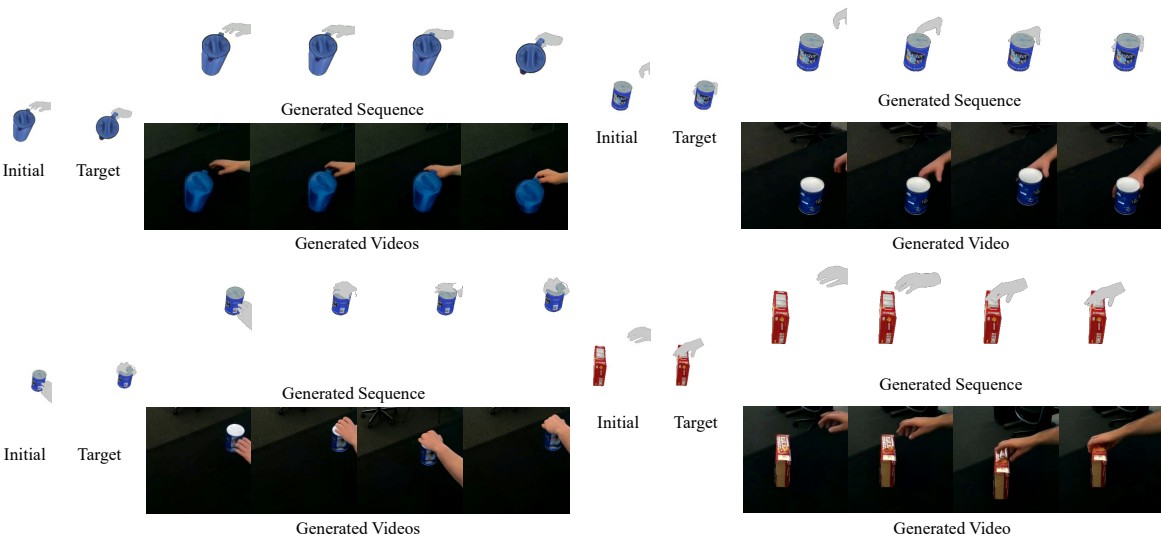

Figure 12: More Sim-to-real transfer results.

