# OpenReview forum: "Sim2Real-HOI: Sim-to-Real HOI Video Generation via Decoupled Motion–Appearance Diffusion"
_ICLR.cc/2026/Conference — ICLR 2026 Conference Withdrawn Submission_

### Official Review · Reviewer_WDQg · 2025-10-30

**Soundness:** 1
**Presentation:** 2
**Contribution:** 1
**Rating:** 2
**Confidence:** 5

**Summary:**

This paper proposes a pipeline for HOI video generation, conditioned on the hand-object pose sequence. Experimental results show that the proposed method achieves better performance on an HOI dataset, i.e., DexYCB.

**Strengths:**

The method is easy to follow. The proposed pipeline achieves better video generation quality on the lab-captured DexYCB dataset.

**Weaknesses:**

1. The technical contributions are severely insufficient. There is neither a newly-designed network architecture nor novel training strategies. The entire pipeline is just a combination of existing works and training on the HOI dataset.

2. The paper claims the "lack of visual diversity" drawback of the first-frame conditioned video generation pipeline in the introduction; however, the proposed pipeline is still based on generating the first frame of the video. I don't think training on a lab-captured dataset, such as the DexYCB dataset, can result in a significantly improved diverse generation ability.

3. I suspect the zero-shot generalization results on the OckInk2 dataset shown in Figure 7. Without training on the OckInk2 dataset, how to make the generated background exactly the same as the ground truth, even for some objects not in the conditions?

4. Improving the performance of a single baseline (SimpleHand) in the downstreaming task is not enough to demonstrate the effectiveness.

 5. The generation ability for more unseen grasping motions should be further analyzed, and more qualitative and quantitative results should be provided.

**Questions:**

Please refer to the weaknesses.

---

### Official Review · Reviewer_QqXY · 2025-11-01

**Soundness:** 3
**Presentation:** 3
**Contribution:** 2
**Rating:** 6
**Confidence:** 2

**Summary:**

This paper proposes Sim2Real-HOI, a framework for hand–object interaction (HOI) video generation from the initial and target hand–object poses, designed to bridge the sim-to-real gap in video generation. Existing HOI video generation methods are less scalable due to their reliance on the availability of the first frame of the video, which introduces (1) an input bottleneck, as such data are difficult to obtain, and (2) a diversity bottleneck, since the first frame is fixed. In contrast, the proposed framework requires only the initial and target HOI poses, effectively mitigating both types of bottlenecks. To achieve this, it disentangles (1) appearance generation, which generates the first frame conditioned on the initial HOI pose, and (2) motion generation, which produces the full sequence of HOI poses. The resulting pose sequence is then fed into a controllable video generation model to synthesize the final HOI video. Experimental results show that the proposed method outperforms prior works in HOI video generation and also improves hand pose estimation performance when the synthesized data are used for additional training.

**Strengths:**

**(1) Strong experimental results**

The proposed framework demonstrates strong experimental performance, outperforming prior works by a noticeable margin in controllable HOI video generation. In addition, the application results (sim-to-real transfer, downstream task validation, and zero-shot evaluation) are well presented, and they clearly demonstrate the practical effectiveness of the proposed method.

**(2) Well-motivated method design**

The method is well motivated, and the rationale behind each component design is clearly justified. The overall motivation is coherent and effectively conveyed.

**(3) Good presentation quality**

Overall, the presentation quality of the paper is very good. It is easy to follow, and the motivation of the proposed method with respect to prior work (L73–80) is particularly well explained.

**Weaknesses:**

**(1) Lack of technical novelty**

Although the proposed method demonstrates strong experimental results and each component design is well justified, the technical novelty is somewhat limited, as it mainly involves adopting existing modules (e.g., ControlNet) and augmenting the conditioning modalities for video generation. Nevertheless, I still recognize the practical value of this work.

**(2) Baseline comparisons for downstream tasks**

The proposed framework could be more convincingly validated if baseline comparisons for the downstream task (pose estimation) were included, as this represents the most practical application scenario.

**Questions:**

L348–349: Could you confirm that the proposed method did not use the ground-truth first frame in the experiments (unlike the baselines)? This clarification is important, as the non-reliance on the ground-truth first frame is a key contribution of the paper.

---

### Official Review · Reviewer_d7o4 · 2025-11-03

**Soundness:** 3
**Presentation:** 3
**Contribution:** 3
**Rating:** 6
**Confidence:** 3

**Summary:**

The paper proposes Sim2Real-HOI for Hand–Object Interaction (HOI) video generation. It aims to bridge the sim-to-real gap using only the initial and target hand–object poses and object geometry as inputs, without relying on the first frame. The approach decouples motion generation and appearance generation. For motion generation, a pretrained hand-motion model generates intermediate poses, which are rendered by a controllable video diffusion model to create full HOI sequences. For appearance generation, a controllable image diffusion model synthesizes the first frame from depth maps, semantic masks, and hand keypoints. The model is evaluated on the DexYCB and OAKINK2 datasets and shows improvements in FVD, SSIM, and MPJPE scores, etc.

**Strengths:**

- The paper is well-written and easy to follow.
- The proposed framework decouples motion and appearance generation, which is simple and intuitive.
- Sim2Real evaluations show that the model trained with synthetic data performs well on real data.

**Weaknesses:**

- The method relies on several pretrained models (e.g., GraspXL, CogVideo-X) rather than an end-to-end training framework. This modular design could introduce cumulative errors between stages. A deeper analysis of these potential issues would strengthen the paper. In particular, the authors should justify their choice of pretrained models and discuss whether adopting more advanced or jointly trained base models could yield further improvements.

- The paper lacks a user study or qualitative evaluation of realism. Including such an evaluation would provide stronger evidence of the model’s real-world applicability and perceptual quality beyond benchmark metrics.

**Questions:**

- In Table 1, Sim2Real-HOI is marked as not using Object Appearance and Hand Appearance as input conditions. However, Figure 1 suggests that the HOI Pose Sequence implicitly reflects both object and hand appearance information. It would be good to clarify how Object Appearance and Hand Appearance are defined in Table 1, and to what extent the pose sequence contributes to appearance modeling. Please correct me if I have misunderstood this aspect.

---

### Official Review · Reviewer_CJT1 · 2025-11-05

**Soundness:** 2
**Presentation:** 3
**Contribution:** 1
**Rating:** 2
**Confidence:** 5

**Summary:**

This paper introduces Sim2Real-HOI for generating HOI videos from initial/target hand and object poses. It proposes a  two-stage process. First, an appearance model synthesizes a photorealistic first frame conditioned on depth, segmentation, hand keypoints. Second, a motion model animates this frame using a pre-trained pose generator and a video diffusion model. This paper demonstrates experimental results on DexYCB and OAKINK2. It also shows the application of using synthetic data to improve downstream hand pose estimation tasks.

**Strengths:**

The paper is well-written and easy to follow. The core ideas are communicated effectively. The proposed cascaded pipeline demonstrates its effectiveness on public datasets, DexYCB and OAKINK.

**Weaknesses:**

1. The paper’s primary motivation focus on overcoming the "first-frame bottleneck" of prior work that conditions on a real first frame. This premise is unconvincing. A realistic first frame can be generated given the initial hand and object pose conditions, with appearance randomization handled by existing image generation models.
2. The proposed pipeline glues a few existing model components together, lacking technical novelty and insights.
3. Relying on a complex, cascaded pipeline of multiple heavy foundation models, including Flux and CogVideoX. This architecture is likely computationally expensive and slow, making it ill-suited for the large-scale data synthesis cited as a goal. The paper lacks a performance study reporting the inference time or computational cost required to generate video clips, which is essential for assessing practical scalability.
4. The experimental validation is restricted to the test splits of the datasets used for training (DexYCB and OAKINK2). The paper lacks in-the-wild evaluation on completely unseen real-world data outside of these established benchmark distributions.

**Questions:**

NA

---

### Note · Authors · 2025-11-14

I have read and agree with the venue's withdrawal policy on behalf of myself and my co-authors.